# In Vitro Glioblastoma Model on a Plate for Localized Drug Release Study from a 3D-Printed Drug-Eluted Hydrogel Mesh

**DOI:** 10.3390/cells13040363

**Published:** 2024-02-19

**Authors:** Behnad Chehri, Kaiwen Liu, Golnaz Vaseghi, Amir Seyfoori, Mohsen Akbari

**Affiliations:** 1Laboratory for Innovations in Microengineering (LiME), Department of Mechanical Engineering, University of Victoria, Victoria, BC V8P 5C2, Canada; chehnad@gmail.com (B.C.); kliu678@student.ubc.ca (K.L.); golnazvaseghi@yahoo.com (G.V.); 2Terasaki Institute for Biomedical Innovation, Los Angeles, CA 90064, USA

**Keywords:** glioblastoma, hydrogel, tumoroid, localized drug release, bioprinting

## Abstract

Glioblastoma multiforme (GBM) is an aggressive type of brain tumor that has limited treatment options. Current standard therapies, including surgery followed by radiotherapy and chemotherapy, are not very effective due to the rapid progression and recurrence of the tumor. Therefore, there is an urgent need for more effective treatments, such as combination therapy and localized drug delivery systems that can reduce systemic side effects. Recently, a handheld printer was developed that can deliver drugs directly to the tumor site. In this study, the feasibility of using this technology for localized co-delivery of temozolomide (TMZ) and deferiprone (DFP) to treat glioblastoma is showcased. A flexible drug-loaded mesh (GlioMesh) loaded with poly (lactic-co-glycolic acid) (PLGA) microparticles is printed, which shows the sustained release of both drugs for up to a month. The effectiveness of the printed drug-eluting mesh in terms of tumor toxicity and invasion inhibition is evaluated using a 3D micro-physiological system on a plate and the formation of GBM tumoroids within the microenvironment. The proposed in vitro model can identify the effective combination doses of TMZ and DFP in a sustained drug delivery platform. Additionally, our approach shows promise in GB therapy by enabling localized delivery of multiple drugs, preventing off-target cytotoxic effects.

## 1. Introduction

One of the most aggressive human cancers is Glioblastoma (GBM), a grade IV brain tumor [1,2]. Less than 6% of patients globally survive longer than five years; the average survival period after diagnosis is 10 to 13 months [3]. The standard treatment for this tumor involves surgical resection followed by chemotherapy (Temozolomide) and radiation therapy [4]. Numerous specific characteristics of the tumor can explain the resistance to treatment and recurrence of GBM. Glioblastoma displays mutational diversity; the deep brain infiltration of glioma (stem) cells precludes the effectiveness of resection; the blood–brain barrier (BBB) limits the access of numerous systemic anticancer drugs to achieve adequate concentrations in the brain. In addition, many GBM recurrences occur within or near the radiation field [5].

These results imply that the resection cavity in surgical site plays a crucial role in preventing early tumor recurrence. Additionally, the blood–brain barrier offers a unique opportunity for localized treatment with fewer systemic side effects [6]. Temozolomide (TMZ) is an FDA-approved chemotherapy alkylating prodrug used to treat GBM by alkylating DNA. Its effectiveness is highly dependent on a strict treatment schedule [7]. TMZ only reaches 30% concentration in the BBB compared to plasma. The current dosage leads to severe side effects such as nausea, vomiting, and lymphopenia. Delivering it locally may help avoid some difficulties associated with TMZ treatment [8,9]. 

However, ineffective drug transport, tumor heterogeneity, and drug resistance pathways have made it difficult for single chemical therapy to demonstrate meaningful improvements for GBM patients. Combining medications could address some of these issues [10]. The best drug combinations work to improve efficacy, reduce toxicity, and reduce doses and drug resistance by utilizing the advantages and disadvantages of each individual molecule [11]. 

Experimental iron chelators have proven to show strong anti-neoplastic effects in vitro in different cancer types [12]. They exert these effects by removing iron from cancer cells and facilitating iron’s redox cycle to produce lethal reactive oxygen species [13]. In brain tumor cell lines, the bidentate iron chelator deferiprone (DFP) exhibits antiproliferative and cytotoxic effects, inhibiting cell growth in a dose and time-dependent manner [14,15]. It has been shown that the combination of TMZ and DFP has a synergistic effect on glioma, and the combination therapy results in a significant G2/M cell cycle arrest [16], and therefore, a DFP and TMZ combination is a potent candidate for GBM treatment.

A possible method for administering greater medication dosages with less deleterious effects on other organs is localized sustained delivery of TMZ and DFP using biodegradable polymeric substrates. Therefore, in this study, we developed a TMZ-DFP topical delivery system based on our previous work [17]. We printed a sustained drug-releasing mesh called GlioMesh that can release an anticancer drug at the tumor site over the course of a specific time. The GlioMesh is constructed of an alginate hydrogel and filled with PLGA microparticles containing TMZ, DFP, or both. In the fabrication process of these microparticles, an oil-in-oil (O/O) emulsion solvent evaporation method was used to generate particles containing TMZ with an external oil phase of liquid paraffin. In contrast, microparticles containing DFP were fabricated using the water-in-oil-in-water (W/O/W) double emulsion method with an aqueous phase of polyvinyl alcohol (PVA). The drug-releasing mesh was created using a micro-extrusion 3D printing technique after the drug-loaded microparticles were suspended in an aqueous alginate solution. Due to larger surface-to-volume ratios, better cellular penetration, and improved supply of nutrients and oxygen to the underlying tissue, the fabrication of a porous mesh via 3D printing can potentially increase drug mass transfer to the surrounding tissue. To this end, we printed a drug-releasing mesh using our recently developed a handheld printer. This handheld bioprinter can print multi-material hydrogels capable of delivering up to four biomaterials including different therapeutic agents to the point of need [18]. The usage of a handheld bioprinter offers a promising approach for personalized cancer therapy by allowing localized drug delivery to the tumor. The current handheld 3D bioprinters can be employed to print therapeutic hydrogels in different geometries and sizes, thereby making it possible to tailor the treatment to fit the local site of the surgery. In the direction of replicating a complex cellular microenvironment which is used for studying the key cellular parameters and drug responses in in vitro experiments, we developed a scalable and open-surface microfluidic-integrated culture plate (MiCP) insert to generate uniform tumor spheroids. This platform can be tailored for different solid tumor models and allows for efficient assay evaluation on the same plate. The MiCP insert builds upon our previous self-filling microwell arrays (SFMAs) technology, which produces an array of uniform size and morphology tumor spheroids using guide channels that distribute cells evenly to the microwell array with the help of gravity [19]. In this study, we used addressable ultra-low-attachment microwells with micro-sized channels to create multicellular tumor spheroids of glioblastoma (GBM) embedded in the extracellular matrix. This model allowed us to evaluate the toxicity of drug-encapsulated GlioMesh on the GBM tumoroids within their microenvironment. By measuring the localized drug release at the site of the tumoroid in the well, we were able to determine the optimal dose of anticancer chemotherapeutic drugs in both single and combination therapy settings. Our MiCP inserts feature microwells that enable the formation of GBM multicellular tumor spheroids in the collagen extracellular matrix for advanced toxicity measurement.

## 2. Materials and Methods

### 2.1. Materials

Agarose was purchased from Bio Basic Inc. (Markham, ON, Canada). Dulbecco’s Modified Eagle Medium (DMEM), fetal bovine serum (FBS), Dulbecco’s phosphate-buffered saline (DPBS), and penicillin/streptomycin (pen/strep) were purchased from Gibco (Grand Island, NY, USA). A 6-well plate was obtained from Corning (Corning, NY, USA). Poly (d, l-lactide-co-glycolide) (PLGA), Temozolomide (TMZ), Deferiprone (DFP), Span 80^®^, and sodium alginate were purchased from Sigma (St. Louis, MO, USA). liquid paraffin was obtained from Caledon Laboratories (Georgetown, ON, Canada)).

### 2.2. Preparation of DFP-Loaded PLGA Microparticles

#### 2.2.1. W/O/W Double Emulsion Solvent Evaporation Method

PLGA microparticles encapsulated with DFP were prepared using a w/o/w method [20]. The first aqueous phase was fabricated using a desired amount of PVA to obtain a concentration of 1% (*w*/*v*). The solution was then placed on a magnetic stirrer for 10 min at a temperature of 85 °C. Afterwards, a desired amount of PLGA was dissolved in 3 mL of dichloromethane and vortex to ensure that PLGA would dissolve completely. Then, 1 mL of DFP with a concentration of 1 mM was added to the oil phase and vortexed for complete dispersion of the drug within the oil phase. The subsequent oil phase was added to the previous PVA solution and mixed using a vortex. The second aqueous phase was prepared by dissolving PVA to obtain 60 mL of 0.2% (*w*/*v*) concentration of PVA. The previously generated solution was added to our second aqueous phase (0.2% PVA). The resulting solution was put on a magnetic stirrer and stirred for 4 h to efficiently generate particles.

The resulting slush was then transferred to a centrifuge and centrifuged at 350× *g* for 5 min. The supernatant was collected and replaced with distilled water to wash the particles, and this process was repeated three times. The final solution was lyophilized for 48 h to ensure no remaining drug or water was left on the surface.

#### 2.2.2. O/O Emulsion Solvent Evaporation Method

An o/o fabrication process was used for DFP-loaded microparticles to achieve the first oil phase, wherein a certain amount of PLGA was dissolved in 3 mL of acetonitrile to generate the desired concentration. A secondary oil phase consisted of 40 mL of liquid paraffin, and 200 µL of span 80. Next, 20 mL of the secondary oil phase was put aside, and the first oil phase was added and vortexed to mix the two oil phases. The resulting solution was then added to the secondary oil phase, placed on a magnetic stirrer, and left to stir for 2 h at a temperature of 55 °C. The final product was transferred to a centrifuge at 350 RCF for 5 min. The resulting particles were then gathered and washed thrice with n-Hexane. After, the final solution was placed under a biosafety cabinet for safe organic solvent evaporation for 48 h or until the particles are completely dry. The particles were dispensed into Eppendorf flasks, and 700 µL of 1 mM DFP were added on top of the particles for 72 h and vortexed twice daily. The final particles were then washed as previously described and freeze-dried for 48 h so that no remaining drug was left on the surface of the particles.

### 2.3. Preparation of TMZ-Loaded PLGA Microparticles Using o/o Emulsion Solvent Evaporation Method

A previous method of o/o emulsion was used by Hossein Zadeh et al. to encapsulate TMZ inside the PLGA microparticles, Figure 1A. 3.75 mg of TMZ was dissolved in 3 mL of acetonitrile and vortexed for 5 min. Then, 200 µL of span 80 was added to 40 mL of liquid paraffin to generate the secondary oil phase. As previously performed, 20 mL of the secondary oil phase was put aside to be mixed with the first oil phase and vortexed. The solution was then added back to our secondary oil phase and placed on a heated magnetic stirrer to let stir for 2 h at 55 °C. The solution is then placed inside a centrifuge and collected. The fabricated particles were washed three times using n-Hexane and left under a hood for 48 h for complete evaporation of the organic solvents (n-Hexane and span 80 and paraffin).

### 2.4. Characterization of PLGA Microparticles

#### 2.4.1. Encapsulation Efficiency (EE) Analysis

To measure the encapsulation efficiency, microparticles generated using the previously described methods were collected at a desired amount of weight (100 mg) and then mixed with the corresponding PLGA solvent (3 mL) by continuous stirring. Dichloromethane was used for microparticles fabricated using the w/o/w method, and acetonitrile for microparticles generated using the o/o method. An amount of 100 µL of supernatant was then extracted after multiple centrifugations at 15,000 rpm for 5 min and placed inside a 96-well plate and then put in a plate reader (Tecan Infinite M200Pro). The plate reader measured DFP and TMZ at 279 nm and 327 nm wavelengths, respectively. The results compared with the previously measured standard curve of the TMZ and DFP in Dichloromethane and acetonitrile. The equation below was used to calculate the encapsulation efficiency [21].
Encapsulation Efficiency (%) = (TMZ or DFP) in supernatant/initial amount of (TMZ or DFP) × 100

#### 2.4.2. Morphology of PLGA Microparticles

Fabricated TMZ and DFP particles with different concentrations of PLGA (5, 10, and 15%) were observed using a scanning electron microscope (SEM) (Hitachi S-4800). To achieve this, 1 mg of each type of categorized particle was mixed with 1 mL of anhydrous ethanol, and subsequently, 10 µL of the solution was placed on SEMstubs and let dry. The dried samples were then placed inside the SEM and were analyzed followed by imaging using open-source software (ImageJ, National Institutes of Health, Bethesda, MD, USA, version 1.54 g).

### 2.5. Three-Dimensional Bioprinting of the Alginate Mesh

To make the drug-eluted alginate mesh, a micro-extrusion handheld bioprinter [18] was used to print the PLGA microparticle included alginate hydrogel. To do this, 9 mL of distilled water was mixed with 1.6 g of alginate to achieve a final concentration of 16% *w*/*v*. Alginate powder was mixed thoroughly overnight to dissolve the alginate completely, Figure 1B. Afterwards, a predetermined amount of PLGA particles (TMZ-encapsulated PLGA particles) depending on the final concentration of particle-to-alginate ratio (1, 3, and 6 mg/mL) were mixed with the alginate solution using two syringes and a connector to mix the particles and alginate. The final alginate mixture was then loaded into the handheld printer cartridges, installed in the core channel of the printhead, and printed using various printing speeds ranging from 100 to 400 mm/min and differentiating printing pressure of 80 to 160 kPa, Figure 1C. The alginate mesh was printed and cross-linked with CaCl_2_ solution while CaCl_2_ was injected through the sheath channel of the printhead during the printing process. The different parameters of the printed mesh were then analyzed using a light microscope (Zeiss Axio Observer, Oberkochen, Germany) and the porous diameter was then measured with the use of ImageJ.

### 2.6. In Vitro Drug Release Experiment

In vitro release studies for both methods of particle fabrication for DFP-loaded and TMZ-encapsulated microparticles were conducted using absorbance measurement of drugs in microparticle supernatant. Furthermore, the same method was used to measure the drug released from the alginate meshes. A known number of particles were placed in Eppendorf tubes, filled with 700 µL of Dulbecco’s modified Eagle medium (DMEM) and artificial cerebral spinal fluid (ACSF) in triplicate, and placed inside an incubator running at 37 °C for specific time stamps. The tubes were gathered and placed in a centrifuge at 15,000 rpm for 5 min to ensure all particles gathered at the bottom. Next, 100 µL of the supernatant was gathered and then replaced with 100 µL fresh media. The solution gathered was then placed inside a plate reader, and the number was measured using the previously mentioned wavelengths (refer to Section 2.4.1).

### 2.7. Brain Tumoroid Formation on a Plate

U251 multicellular tumor spheroid formation was performed using the hydrogel-based microfluidic-integrated culture plate from Apricell biotechnology (3-in-1 PLATE) with ultra-low-attachment microwells, Figure 1E,F. Hydrogel inserts were fabricated by replica molding of the 3D-printed microfeatures. A 3-in-1 PLATE^TM^ insert inside the six-well plates was seeded gently with 100 µL of culture media, including 2 × 10^5^ cells on a seeding zone of the plate. The loaded device was incubated at 37 °C for 10 min to let the cells fill the microwells through the microchannels of the device. Afterward, 300 µL of fresh culture media was gently added inside the media reservoir of the plate to cover the cells in microwells. U251 spheroids were monitored daily to measure their growth over the three days.

### 2.8. ECM Embedding and Combinatorial Drug Testing on Brain Tumoroid Models

Regarding the open-surface design of the microfluidic-integrated culture plate, brain tumoroids were easily embedded inside the ECM hydrogel without the need for any further manipulation. In order to replicate the extracellular matrix (ECM) conditions found in glioblastoma tumors, bovine fibril collagen was used as the main component of the tumor ECM, with a concentration of 4 mg/mL. For the hydrogel solution preparation, the pH and ionic concentrations of the collagen solution were adjusted by combining 10× PBS and 0.5 N NaOH with the collagen stock solution at a 1:1:8 ratio. Subsequently, a specific volume of culture media was promptly mixed with the collagen solution to attain the desired concentration. The final collagen solution was gently introduced into the ECM loading zone of the culture plate insert and then incubated at 37 °C in an incubator for 45 min to achieve complete cross-linking. To investigate the effect of iron-chelating drugs on the metabolic activity and cytotoxicity of the brain tumoroids in a single and combination therapies with TMZ, tumoroid drug testing was conducted with DFP and various concentrations of TMZ microparticle-loaded alginate mesh (1–6 mg/mL) in a single and combination treatment setting. In this method, without removing the tumoroids from the microwell, the 4-day-old tumoroids were treated with the iron chelator drug by aspirating the primary culture medium from the media reservoir followed by exposing the tumoroids to the fresh 300 μL of DFP microparticle resale media with the concentration of 100 µM for one day. In a combinational drug treatment setting, alginate meshes with different concentrations of TMZ microparticles were directly printed and inserted inside the media reservoir of the tumoroid culture plate for localized drug release study for 1 and 5 days following iron-chelating drug treatment on tumoroids on a plate, Figure 1E. To compare the effectiveness of combinatorial DFP/TMZ treatment, a control group was established where the tumoroids were only treated with TMZ microparticle incorporated alginate mesh. The tumoroids’ viability and invasion behavior were measured at different time intervals after the DFP treatment. Furthermore, the viability and invasion behavior were measured on day 1 and 5 after the DFP/TMZ treatment. Figure 1G–I illustrate the invasion process of glioblastoma tumoroids within the ECM hydrogel over time.

### 2.9. Drug Toxicity and Live/Dead Staining and Invasion Assay of the Tumoroids on a Plate

The viability of cancer cells within tumoroids was examined using a live/dead assay. This involved using 1 μM calcein AM and 4 μM ethidium homodimer-1 from a Life Technologies kit for 30 min at 37 °C. The entire ECM-embedded GBM tumoroid model was stained and imaged on the culture plate insert without removing the tumoroids before staining. Fluorescent imaging and quantification of invaded cells with the ECM assessed the effect of drug doses on single and combination treatment interventions. To determine the amount of drug toxicity for each concentration and day of drug application on the tumoroid cells, a Presto Blue assay was used. This assay measures the live cells’ metabolic activity using the fluorescence excitation and emission wavelengths of 560 nm and 590 nm, respectively. The Presto Blue reagent, which is 10% of the total culture medium, was added directly to the media reservoir of the insert, covering the tumoroids during the 3 h incubation. The average fluorescence intensity of the tumoroid was subtracted from the intensity of related cell-free microwells in the presence of the blank sample. The length of invasion of tumoroids in the collagen ECM was measured using fluorescence microscopy on a plate. We used image analysis software called ImageJ, version 1.54g to normalize the invasion length to the initial size of the tumoroids after they were embedded in the ECM. This helped us to obtain standard and comparable results for the invasion behavior of tumoroids under different treatment conditions.

### 2.10. Flow Cytometry Analysis

To quantify the population of the viable and dead cells within the tumoroids under different treatment conditions, we conducted the flow cytometry analysis (Attune NxT, ThermoFisher Scientific, Waltham, MA, USA). To prepare the samples for flow cytometry, tumoroids were dissociated using a mixture of trypsin 1X and collagenase type I (0.001 mg/mL) for 5 min in an incubator. Dissociated single cells were removed from the reservoir of the plate and prepared for staining. For flow cytometry staining, 2 µL of 50 × 10^−6^ M calcein solution (in DMSO) and 4 µL of 2 × 10^−3^ M ethidium homodimer-1 were introduced into 1 mL of cell suspension and incubated in a dark condition at room temperature for 20 min. The excitation wavelength was set at 488 nm, and the emission filters were 530/30 nm bandpass and 610/20 nm bandpass for calcein AM (green) and ethidium homodimer (red), respectively. To eliminate debris from the resulting dot plot, appropriate gating was applied to distinguish live and dead cell populations.

### 2.11. Statistical Analysis

Each experiment was conducted with a minimum of three replicates, and the outcomes were expressed as mean ± standard deviation (SD). Statistical comparisons of the means were performed using one-way ANOVA with Tukey’s multiple comparisons test in GraphPad Prism 7.0. Significance levels were denoted as follows: * *p* < 0.05, ** *p* < 0.01, *** *p* < 0.001, and **** *p* < 0.0001.

## 3. Results and Discussions

### 3.1. DFP-Loaded and TMZ-Encapsulated Microparticles

PLGA particles have been widely researched as an alternate method to administer various anticancer drugs systematically. PLGA is used for its wide range of capabilities, such as biocompatibility and the ability to be able to release drugs through diffusion sustainably [22]. Here, we studied different fabrication methods for DFP-loaded microparticle fabrication to determine the optimal fabrication method using different parameters. Table 1 depicts the encapsulation efficiency of different particles using various methods and PLGA concentrations used for particle fabrication.

The results indicate a low encapsulation efficiency for the o/o method, while the encapsulation efficacy of DFP is reported to be higher for the w/o/w method due to its water-soluble behavior [23]. The w/o/w method has been proven to be superior and produce a higher DFP encapsulation efficacy. However, the encapsulation rate is dependent on the concentration of PLGA in the oil phase for both methods. As the concentration of PLGA increases, a higher matrix network is created, which physically traps the drug. Therefore, loading, as compared to the in situ encapsulation method, which was previously used, would not yield beneficial results.

For making TMZ-encapsulated microparticles, a previously reported protocol by Hosseinzadeh et al. [17] was used. Due to the hydrophobic nature of TMZ, the o/o method was used for the fabrication process by varying the PLGA concentrations. Similarly, to the methods used for DFP, the increase in PLGA concentration from 5% to 15% (*v*/*w*) directly impacted the encapsulation efficacity increase in PLGA, resulting in the slower diffusion of the drug.

### 3.2. Physical Characterization of DFP- and TMZ-Encapsulated PLGA Microparticles

Fabricated PLGA microparticles with different polymer concentrations were characterized by scanning electron microscopy (SEM) using a Hitachi S-4800 microscope. Figure 2 shows the particles created using the o/o method and w/o/w method for TMZ and DFP, respectively, while varying the PLGA concentration. The SEM images depict that for the TMZ-loaded microparticles, there is more of a homogenous distribution of the particles, with the size increasing as the PLGA concentration increases. The microparticles loaded with DFP exhibit a more diverse range of sizes among the different PLGA concentrations, resulting in a non-uniform particle distribution. This may reflect the role of DFP as an additive to the aqueous phase of the double emulsion system, which establishes a direct correlation between the PLGA concentration and particle size, as evidenced in Figure 2A–C. As the PLGA concentration increases, the average size of the microparticles increases. This trend is more pronounced in the DFP-encapsulated microparticles as compared to the TMZ-encapsulated ones. This could be attributed to the effect of increased viscosity resulting from higher PLGA concentration during the synthesis process [24].

### 3.3. In Vitro Release Assay from the Drug-Loaded Microparticles

Diffusion, hydrolytic breakdown, or combining the two processes leads to the release process from PLGA microparticles [25]. According to the literature, hydrolytic breakdown mechanism in PLGA microparticles is size-dependent, so that PLGA microparticles degrade heterogeneously and more quickly when they are significantly bigger than 300 μm [26]. In this study, we researched both fabrication methods of DFP to determine their effectiveness in releasing the drug. As shown in Appendix A, we observed an initial burst release of the drug in the DFP particles fabricated using the o/o method, which is consistent with previous research. However, the cumulative release of the drug is relatively low, which can be attributed to poor encapsulation efficiency. During organic solvent evaporation, DFP may diffuse into the oil phase stabilizer solution, resulting in a lower EE compared to the w/o/w method, as observed in the results [21]. Moreover, the release profile shows a steady release with little signs of a plateau, meaning the drug is still being released. Similarly, as shown in Figure 3A,B, we see the same trend depict itself in the release of DFP from particles fabricated with the w/o/w method. We notified that the initial burst has reappeared, but the latest fabrication method shows more concentration of drug being released inside different media. In contrast to the TMZ microparticles shown in Figure 3C,D, there is no initial burst release of the drug. This difference can be attributed to several factors. Firstly, the rate of degradation of the particles is highly influenced by the media used (DMEM and ACSF), which significantly affects the rate at which the drug is released from the particles. It is important to note that the drug release time frames for DFP and TMZ were selected differently. DFP, being an iron chelator agent, needs to have an immediate effect on the metabolism of tumor cells and make them sensitive to TMZ treatment in a short time frame. According to the literature, iron chelators should only be applied for a short period of time to induce upregulation of Hif1-α expression. This results in tumor-suppressing effects by causing iron depletion and making the GBM more sensitive to TMZ treatment [27].

### 3.4. In Vitro Release Assay from the Drug-Loaded Microparticles in 3D-Printed Alginate Mesh

Figure 4 depicts photographic images of the alginate mesh and illustrates how differentiation variables such as printing speed and printing pressure affect the final alginate substrate. The results depict a correlation between the varying pressure, printing speeds, particle concentration, alginate mesh fiber diameter, and porous size. Our results indicate that as the printing pressure increased from 80 kPa to 160 kPa, the printed fiber diameter decreased due to an increase in bioink (alginate) deposition at the printing site. After conducting experiments with varying printing speeds, we observed that the printing speed has an opposite effect on the diameter of the printed material. Specifically, increasing the printing speed from 150 mm/min to 350 mm/min resulted in less alginate being printed, and hence, a decrease in diameter. We selected the printing speed of 250 mm/min at 120 Kpa as the optimal printing parameter for printing the mesh with different TMZ microparticle concentrations in the subsequent step. Moreover, as shown in Figure 4G–I, the printed alginate mesh was also characterized when the particle-to-alginate ratio was changed from 1 mg/mL to 6 mg/mL. These changes indicate that an increase in particle concentration leads to an increase in printed fiber diameter, similarly to the effect seen when the printing pressure is changed. As the ratio of particles to bioink (alginate) increases, the bioink becomes more viscous, which affects the speed at which the alginate can be printed. Additionally, it should be noted that placing more particles in the bioink causes the printing syringe to clog, resulting in lower mesh quality.

Utilizing a localized drug delivery approach can help overcome some of the challenges associated with the standard chemotherapeutic approach for glioblastoma (GBM). This method involves delivering the drug directly to the tumor site, bypassing the blood–brain barrier (BBB). As a result, higher drug doses can be achieved and sustained in the targeted area [28]. As a result, the localized delivery approach can enhance the effectiveness of established and novel drugs in GBM therapy while minimizing side effects on other organs compared to systemic administration. This has the potential to address significant obstacles in the effective treatment of GBM [29]. Figure 4J,K display the release characteristics of TMZ particles in the alginate mesh. The release profile of TMZ from the microparticles within the 3D-printed alginate mesh demonstrates a trend of small amounts of drugs being released without any significant initial burst release of the drug. The initial burst is minimal, occurring only in the first few hours of the study. During the first 48 h of the measuring time, there is a slight decrease in the initial release rate of TMZ in both DMEM and ACSF. However, the final cumulative release of TMZ from the alginate mesh in DMEM is higher than in ACSF. This suggests that the type of media used affects the rate of release, resulting in a slower release in ACSF compared to DMEM. The release profile is dependent on the type of media used, and this could be due to the difference in ionic composition of the release media, which can cause variations in the degradation behavior of the alginate and the release of the drug to the media. According to the literature, DMEM is ionically more complex than ACSF, with a higher concentration of monovalent cations [30]. Ionically cross-linked alginate with divalent ions can be dissolved more quickly in DMEM due to exchange reactions with monovalent cations in the surrounding media [31]. The increases in the particle-to-alginate ratios had little impact on the rate at which TMZ was released, as seen in Figure 4J,K. As explained in the Materials and Methods section, the high concentration of alginate (16% *w*/*v*) used in the experiment led to a delay in the release of TMZ in comparison to the release profile of the TMZ-encapsulated microparticles in the DMEM solution. This behavior can be associated with the barrier effect of the alginate matrix which slows down the release of the TMZ form the 3D-printed mesh structure [32]. Additionally, the data indicates that the release of TMZ from the particles was not significantly affected by the ratio of particle to alginate. In comparison to release profiles of the particles without alginate, the data did show a steady slope during each measurement. These results confirmed that the use of alginate also restricted the release rate, which has been reported in other studies [21]. This was more pronounced in ACSF, and the release rate of TMZ was lower than that of DMEM, as was mentioned in earlier sections. The release profile of TMZ-encapsulated particles made with 5% PLGA concentration is depicted in the figure and was largely smooth with a little burst. It has been observed that the different particle-to-alginate ratios have an impact on the release of TMZ, although this impact is not significant. This supports the theory that the porosity of alginate leads to the release of TMZ from the particles that are trapped in the fibers, resulting in a smooth slope throughout the time points of observation.

### 3.5. In Vitro Tumoroid Toxicity Assay

Bioengineered tumor models, particularly tumoroid-based in vitro models, have shown promise in evaluating the efficacy of current anticancer drugs and the development of new therapies [33,34]. Tumor spheroids have the potential to create higher-order cellular tissue organization and better replicate the real tissue microenvironment [35]. The current hydrogel-based microwell arrays have limitations in creating a comprehensive environment that can fully replicate all the different compartments of the tumor microenvironment, including the ECM and stromal compartments [19]. This means that we need to develop a new platform that can overcome these limitations and provide a more complete modeling of the tumor microenvironment. To address these concerns, we have utilized a new platform called tumoroid on a plate (ToP) which enables the creation of 3D tumor spheroids in standard well plates, which can be used to study important cellular parameters and drug responses, as well as the replication of complex cellular microenvironments. In this study, we mimicked the complexity of the glioblastoma tumor microenvironment using a 3-in-1 tumoroid culture plate from Apricell Biotechnology Inc. (Victoria, BC, Canada), culturing U251 cells in a 3D format, and embedding the tumoroids in a collagen hydrogel to represent the brain tumor microenvironment ECM. We used a 3D culture plate with an open-surface design to test the toxicity of tumoroids against varying concentrations of a drug-eluted 3D printable hydrogel. The TMZ-eluted alginate mesh was directly printed into the reservoir of this culture plate insert (Figure 1E) to investigate the in vitro cellular behavior in response to the locally released therapeutic compound from the 3D-printed mesh. This design allowed us to conduct a systematic viability study that would not have been possible with the existing organs-on-a-chip or traditional 3D spheroid plates. We found this 3D culture plate to be advantageous and demonstrated its feasibility for drug studies. Our experiments involved localized exposure of the tumoroids to various doses of TMZ-encapsulated meshes in the 3D culture plate for five days. To evaluate the effect of localized chemotherapy on the integrity of the tumoroids in our 3D model, we stained the tumoroids on a plate using the live/dead kit. The fluorescent images of the five-day-old treated tumoroids under different treatment conditions are shown in Figure 5A(i). An increased concentration of loaded microparticles inside the alginate mesh resulted in a higher population of dead cells at the core of the tumoroid and increased shrinkage in their size. However, the cells in the treated tumoroids with TMZ meshes were loose and showed some disintegrated single cells from the body of the tumoroids.

The tumor response to different localized drug treatment conditions was assessed through tumoroid viability and invasion assays inside the collagen ECM. The impact of iron chelator free drug on the survival of U251 tumoroids was analyzed and is presented in Appendix A. The figure shows that when the tumoroids were treated with DFP at concentrations ranging from 10 to 100 µM, there was a slight reduction in their viability. The viability in this experiment reflects the metabolic activity of the tumoroids (conducted by Presto-Blue assay) exposed to DFP and varying concentrations of TMZ in different treatment conditions. In Figure 5C,D, it can be observed that the viability of tumor cells was not significantly affected by DFP treatment on day one. However, on day 5 of DFP treatment, the viability of tumoroids appeared to be lower compared to the control condition. On the other hand, when the alginate mesh was 3D printed and loaded with TMZ microparticles at a concentration of 6 mg/mL, there was a gradual increase in cell death observed on day 1 of the treatment, compared to the control conditions and DFP alone treatment. There was no notable difference in the percentage of tumor cell death between the TMZ-only treatment and the combination of DFP and TMZ with 3 mg/mL of TMZ microparticles. However, a significant difference in cell viability was observed between the TMZ alone and the DFP/TMZ treatment conditions with the same dosage of TMZ microparticles (6 mg/mL) in the 3D-printed mesh. This behavior demonstrates the synergistic effect of the combinatorial treatment on the toxicity of the GBM tumor cells in our in vitro model. Consequently, the tumor model showed higher cell toxicity when exposed to TMZ-encapsulated meshes on day five as compared to day one. Additionally, an increasing number of drug-loaded microparticles inside the alginate mesh resulted in a significant cell death percentage as compared to the DFP alone and TMZ alone treatments on day 5. These results are in alignment with flow cytometry results following dissociating spheroid and staining them with a live/dead kit on day 5 of the treatment, Figure 5E–J.

One of the main advantages that our 3D in vitro model offers over conventional 2D culture plates is its ability to model and quantify cell invasion in a 3D cell culture setting [36]. One advantage of the current design is its ability to create tumoroids in a compartmentalized array of microcavities that are connected to a microchannel. This allows for in-site invasion assays on the tumoroids after embedding in the ECM. This capability is not available in current scaffold-based or scaffold-free 3D culture tools on the market, such as Organogenix and Corning microplates, respectively [36]. Several studies indicate that remodeling the 3D cell culture microenvironment can assist tumor spheroids in regulating the effects of therapeutic agents. Additionally, this highlights the importance of cell–matrix interactions that lead to invasion and malignancy [37]. In this study, the invasion was introduced as a marker for evaluating the efficacy of the localized drug treatment. For this purpose, 3D tumor models were formed by encapsulating the formed U251 tumoroids inside the collagen ECM as the main component of the brain tumor microenvironment. As explained in the materials and methods section, tumoroids in ECM were locally exposed to TMZ (1, 3, and 6 mg/mL of loaded microparticles) following DFP treatment for five days. The invasion length of the tumoroids within the ECM in response to various treatment conditions was monitored and quantified through fluorescent microscopy of the stained cells inside the ECM hydrogel and using ImageJ software (1.54g) on day 5, Figure 6F.

As illustrated in Figure 6G,H, there is a significant decrease in the invasion length of the treated tumoroids with DFP and TMZ alone treatment in comparison to the control sample. However, in the combination treatment setting, 1 mg/mL microparticle-loaded mesh did not show any significant effect on the invasive behavior of the U251 tumoroids within the gel compared to the TMZ alone condition. By increasing the number of microparticles loaded in the 3D-printed mesh, the length of tumoroid invasion was significantly reduced. At a concentration of 6 mg/mL of TMZ microparticle combined with DFP, we observed an average invasion length of 25%, which is less than half of the normalized invasion length of tumoroids treated with TMZ alone (6 mg/mL).

## 4. Conclusions

The proposed technique to fabricate microparticles loaded with TMZ and DFP in this study has demonstrated high encapsulation efficiencies and enabled affordable production. These microparticles can be embedded within the 3D-printed GlioMesh using a handheld 3D printing process, allowing for their immobilization at the tumor site. Continuous release of DFP and TMZ from these microparticles has been shown to improve the therapeutic efficacy of the treatment while reducing the risk of systemic toxicity. Handheld bioprinting provides the flexibility to create complex structures and geometries that can be customized to the specific requirements of the patient and the tumor site. Bioprinting can also improve the targeting and localization of the drug-loaded microparticles, resulting in more effective treatment. DFP and TMZ encapsulated in the PLGA microparticles have demonstrated a formulation-dependent release profile. By increasing the concentration of PLGA, the less accumulative release was reported in both DMEM and ACSF media. Additionally, TMZ-encapsulated microparticles within the 3D-printed alginate mesh showed a sustained release profile in both ACSF and DMEM release media. Therefore, the accumulative release increased by increasing the concentration of TMZ-loaded microparticles in the mesh structure. A novel 3D in vitro tumoroid on-a-plate model was implemented in this study to recapitulate the microphysiological behavior of glioblastoma. Three-dimensional tumoroids embedded within their ECM were used to screen the effect of sustained drug release. The toxicity results of the tumoroids were aligned with the release profile of the TMZ-loaded meshes, and it showed higher cancer cell toxicity and less invasive behavior by increasing the concentrations of drug-loaded microparticles in the GlioMesh structure in the co-treatment setting. Overall, the combination of TMZ and DFP with a handheld 3D-printed mesh is a promising approach for enhancing the synergistic effects and improving efficacy. However, additional models should be employed to further evaluate the clinical potential of TMZ and DFP GlioMesh.

## Figures and Tables

**Figure 1 cells-13-00363-f001:**
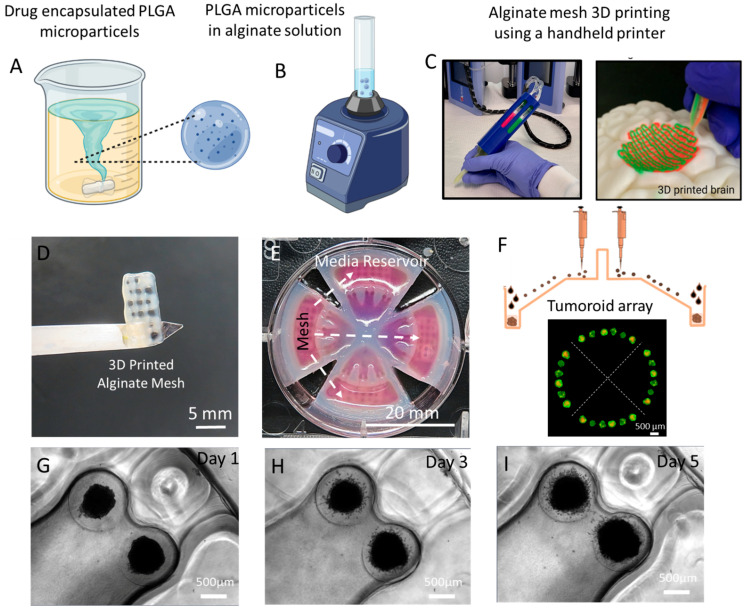
Schematic view of the (**A**) Emulsion solvent evaporation technique for fabrication of TMZ-loaded PLGA microparticles with high encapsulation efficiency and (**B**) bioink preparation. (**C**) 3D printing of alginate mesh containing TMZ-encapsulated PLGA microparticles using a handheld printer. (**D**) 3D-printed alginate mesh, including drug-encapsulated microparticles. (**E**) 3-in-1 3D culture plate with microwells and inserted 3D-printed mesh for localized drug release study. (**F**) Schematic side view of the 3D tumoroid culture plate with the pattern of formed spheroids inside the microwells. (**G**–**I**) Invasion process illustration of glioblastoma tumoroids within the collagen ECM hydrogel over time.

**Figure 2 cells-13-00363-f002:**
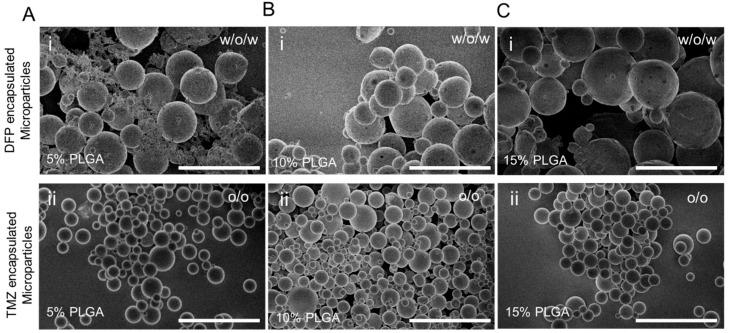
SEM micrographs of various synthesized microparticles. (**A**) 5% PLGA synthesized microparticles encapsulated with (i) DFP and (ii) TMZ. (**B**) 10% PLGA synthesized microparticles encapsulated with (i) DFP and (ii) TMZ. (**C**) 15% PLGA synthesized microparticles encapsulated with (i) DFP and (ii) TMZ. The scale bar is 300 µm.

**Figure 3 cells-13-00363-f003:**
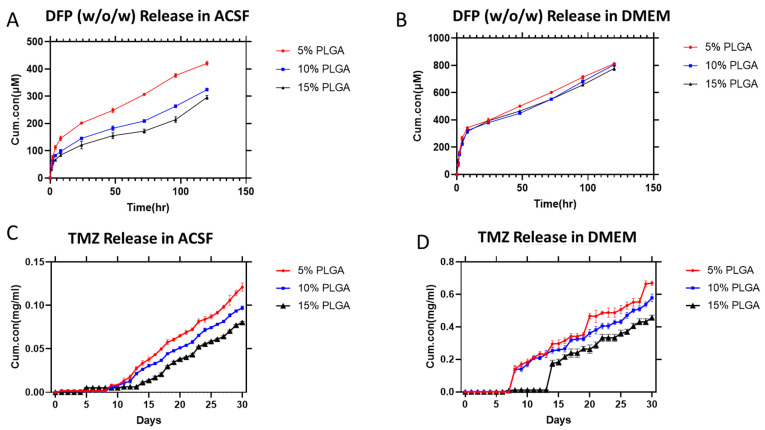
(**A**) Overview of various release profiles of DFP for various PLGA concentrations in ACSF. (**B**) Overview of various release profiles of DFP for various PLGA concentrations in DMEM fabricated with the w/o/w method. (**C**) Overview of various release profiles of TMZ for various PLGA concentrations in ACSF. (**D**) Overview of various release profiles of TMZ for various PLGA concentrations in DMEM fabricated with the o/o method. Cum.con: Cumulative concentration.

**Figure 4 cells-13-00363-f004:**
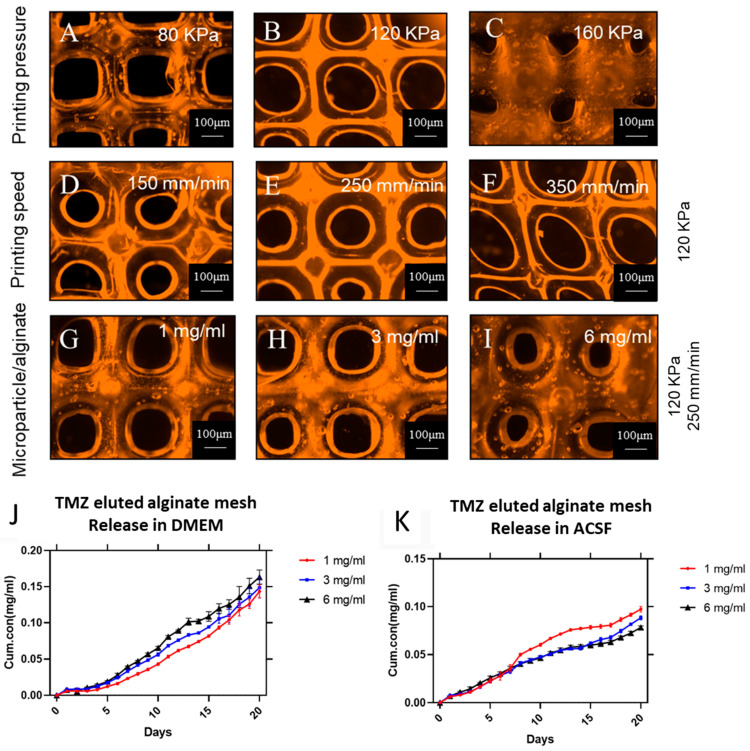
(**A**–**I**) Micrographs of alginate meshes printed using various particle concentrations (1, 3, and 6 mg/mL) and under different printing parameters. (**J**) Overview of various release profiles of TMZ for various alginate-to-particle ratios in DMEM. (**K**) Overview of various release profiles of TMZ for various alginate-to-particle ratios in ACSF.

**Figure 5 cells-13-00363-f005:**
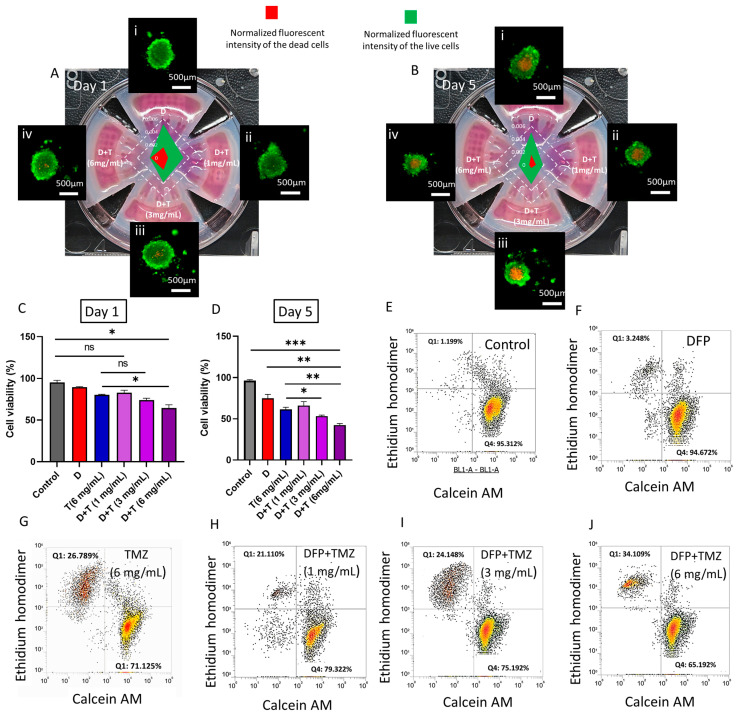
(**A**,**B**) Live/dead imaging of the tumoroids with different treatment conditions on days 1 and 5, respectively. (i) DFP, and DFP + TMZ treatment with different microparticle loading: (ii) 1 mg/mL, (iii) 3 mg/mL, (iv) 6 mg/mL. Normalized fluorescent intensity of the live and dead cell population in tumoroids under different treatment conditions is measured through ImageJ analysis and compared on day one and day five, depicted in panels (**A**,**B**), respectively. (**C**,**D**) Quantitative cell viability of the tumoroids screened with different treatment conditions using presto-blue assay (D: DFP and D + T: DFP + TMZ with varying concentrations inside the 3D-printed mesh) on day 1 and 5, respectively. * *p* < 0.05, ** *p* < 0.01, *** *p* < 0.001, and ns = not significant. (**E**–**J**) Flow cytometry live/dead analysis of the population of cells within treated tumoroids with different GlioMesh conditions on day 5.

**Figure 6 cells-13-00363-f006:**
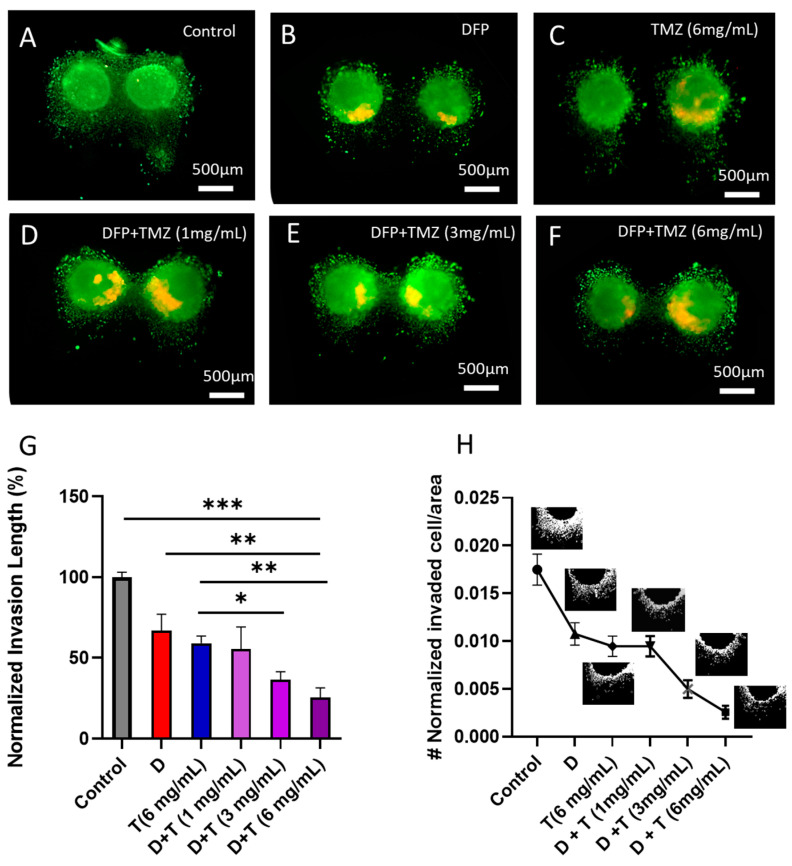
(**A**–**F**) Live/dead imaging of the tumoroids invasion with different treatment conditions on day 5. (**F**) Quantified normalized invasion length of tumoroids under different treatment conditions on day 5 (D: DFP; T: TMZ microparticles; D + T: DFP + TMZ microparticles with varying concentrations inside the 3D-printed mesh). (**G**) Normalized invasion length of the tumoroids in different treatment conditions on day 5; * *p* < 0.05, ** *p* < 0.01, and *** *p* < 0.001. (**H**) Normalized number of invaded cells/areas in different treatment conditions of the U251 tumoroids.

**Table 1 cells-13-00363-t001:** Overview of DFP- and TMZ-loaded microparticles using various methods of fabrication based on their encapsulation efficiencies.

Drug Encapsulated	Fabrication Method	PLGA Concentration	Encapsulation Efficiency
DFP	o/o	5%	11.23%
	o/o	10%	24.57%
	o/o	15%	28.65%
	w/o/w	5%	36.54%
	w/o/w	10%	46.51%
	w/o/w	15%	62.39%
TMZ	o/o	5%	15.52%
	o/o	10%	26.89%
	o/o	15%	31.87%

## Data Availability

Data are contained within the article and Appendix A.

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
