# Peer review of "In Vitro Glioblastoma Model on a Plate for Localized Drug Release Study from a 3D-Printed Drug-Eluted Hydrogel Mesh"

_cells, 2024, doi:10.3390/cells13040363_

Round 1
Reviewer 1 Report
Comments and Suggestions for Authors
In this study, the researchers addressed the limited treatment options for glioblastoma multiforme (GBM), an aggressive brain tumor, by exploring a novel approach using a handheld printer for localized drug delivery. They developed a flexible drug-loaded mesh (GlioMesh) containing poly (lactic-co-glycolic acid) (PLGA) microspheres, showcasing the sustained release of temozolomide and deferiprone for up to a month.
However, the manuscript, in its current form, is not suitable for publication. It is necessary to check the entire paper, repeat some analyses, and interpret them correctly, as well as complete additional experiments.
Some key points that need improvement include:
The entire Section 3, Results and Discussion, contains only 5 references. It is necessary to discuss in detail all results and compare them with similar results from other articles. Highlight the advantages and disadvantages of the study in relation to already existing results that use similar technology or methods of tumor treatment. Also, emphasize the novelty that sets this work apart from already published results.
In the Results section, some experiments are not clearly described, and key controls are missing.
For instance, on line 373, the authors state, "Our experiments involved localized exposure of the tumoroids to various doses of TMZ encapsulated meshes in the 3D culture plate for five days." However, Figure 5 indicates a combined treatment (D+T). Additionally, a key control is missing – the effect of treatment with only TMZ. It is necessary to compare drug combinations with the effect of each drug and concentration individually. Here, the effect of treatment with only TMZ is missing, so that it would be possible to compare and conclude what effect combined treatments have.
In Sections 3.2 and 3.3, there is a lack of control. It is necessary to include PLGA microparticles made without the presence of drugs as controls during physical characterization and in the In vitro Release Assay.
In Section 3.3, where the cumulative concentration of DFP and TMZ was measured in different time frames, it is apparent that the release dynamics of these two drugs in the medium are significantly different. It is unclear how combined treatment will be applied when such a difference exists. It is necessary to emphasize and discuss this difference in detail in the paper.
In Section 3.4, it is necessary to examine the release and cumulative concentrations of DFP. Additionally, during the production of the alginate mesh, it is not clear which pressure was used when the speed was varied, as well as the printing speed when the pressure was varied. It was not stated which speed and pressure were finally chosen for further experiments.
Author Response
"Please see the attachment."

Reviewer 2 Report
Comments and Suggestions for Authors
In the manuscript, the authors presented applicative new technology research – they used the newly developed handheld printer in assessing temozolomide and deferiprone drugs localized co-delivery in glioblastoma tumoroids. The manuscript is well written; the results are properly graphically presented and described; the discussion and conclusions are supported by the results.
Strongholds: The research is highly applicable in different modalities and presents a glimpse of the potential it holds.
Downsides: The commercial cell line (U251) holds very limited translational potential to a GBM tumor. Spheroids composed of closely to identical cells do not reflect the heterogeneity of GBM cancer cells in the tumor. Nevertheless, this is a sound methodological postulate and can be further developed in the future.
Other comments are as follows:
1. Line 30: the facts mentioned in the statement do not match the reference.
3. Line 42: TMZ is a prodrug.
4. Line 54: Malignancy is not an appropriate word for in vitro models.
5. Line 132: rcf or rpm?
6. The dose release assessment should be described in more detail and appropriate references cited (section 2.4.1).
7. Line 234: is m referring to molar concentration? If so, the unit should be capitalized M.
8. Line 235: what is 1 mL -1 of cell suspension?
9. In section 2.8 please specify the details of the treatment – total duration, measurements, schedules, etc.
10. In section 2.9 please add details on how you prepared/dissociated the spheroids for flow cytometry.
11. Please add in Figure legends what cum.con is (Figure 3).
12. Regarding the differences in drug release in DMEM vs. aCSF – do you have a hypothesis what is the source of differences observed?
13. Line 325: Please check the sentence starting with “Our results show…”.
14. Line 330: Please check for clarity the sentence starting with “Moreover, alginate mesh…”.
15. Figure 4A-I – the unit for pressure is kPa – you’ve written it correctly at some parts of the manuscript, but in Figure 4 and in line 326 correction should be made.
16. Please add appropriate references for statements in section 3.5: “Bioengineered tumor models, particularly tumoroid-based in-vitro models, have shown promise in evaluating the efficacy of current anticancer drugs and the development of new therapies. Tumor spheroids have the potential to create higher-order cellular tissue organization and better replicate the real tissue microenvironment.”
17. In Figure 5A&B – Please specify how you quantified cell viability in the Figure legend and/or the Materials and Methods. The same goes for Figure 6 – please describe how you quantified the normalized invasion length.
Comments on the Quality of English Language
In vitro, without the dash.
Lines 43 and 57: there are extra dots.
Line 83: The space is missing in “in vitro”.
Line 90: an addressable.
Line 128: typo.
Line 193: typos
Line 236: “dark light” is an oxymoron; “under dark conditions/in absence of light” is a suggestion.
Author Response
"Please see the attachment."

Reviewer 3 Report
Comments and Suggestions for Authors
Chehri et all.
In-Vitro Glioblastoma Model on-a-Plate for Localized Drug Release Study from a 3D-Printed Drug-Eluted Hydrogel Mesh
In this study, the feasibility of using this technology for localized co-delivery of temozolomide (TMZ) and deferiprone (DFP) to treat glioblastoma was tested. Authors showed that that 3D tumoroids derived from conventional glioblastoma cell line (U251) respond to combinatorial effect of TMZ and DFP. In this study authors hypothesized that DFP will potentiate effect of TMZ.
Critiques:
In the introduction authors set rationale to test combinatorial effect of DFP and TMZ release aimed to increase efficiency of cell killing. However, in vitro cell viability in Fig 5 and inhibition of cell invasion, fig.6, are measured after DFP only treatment and increased dosing of DFP plus TMZ microspheres imbedded in the printed mesh. No experiments with TMZ only treatment were performed. No potentiation of DNA damage (or other possible molecular events) induced by TMZ in the presence of DFP as a possible mechanism were investigated. As a result it remains unclear if DFP potentiates effect of TMZ. Therefore, the conclusion that DFP potentiate TMZ response is not supported by experimental results.
Authors aimed to apply previously developed drug treatment technology for testing drug treatment using 3D culture. The manuscript will be significantly improved if primary glioma cells derived from patients will be used in serum free conditions. Because of a possible variability of glioma cell intrinsic response to TMZ, application of multiple cell lines/primary cells will support author’s conclusion.
Minor critiques:
Because authors previously published in vitro treatment technology (ref 16) the detailed repetition of the mesh synthesis process description is not needed unless authors introduced substantial differences. If so, main emphasis in technology description should be given describing any improvement/modifications.
Result discussion is virtually absent. Authors could include discussion application of technology for drug screening, personalized medicine, compare with nanotechnology in the context of 2 and 3D culture, including tumor slice culture, and in vivo application.
Author Response
"Please see the attachment."

Round 2
Reviewer 1 Report
Comments and Suggestions for Authors
The authors significantly improved the article after revision.
Author Response
Thank you for your constructive comments.
Reviewer 3 Report
Comments and Suggestions for Authors
Authors responded to critique.
Minor: for Fig.5 E-J, please indicate time of treatment.
Thank you.
Author Response
Thanks for this comment.
The day of treatment was mentioned in the caption of Figure 5 and the manuscript.